# Earthquake focal mechanisms with distributed acoustic sensing

Jiaxuan Li [1] ✉, Weiqiang Zhu[1], Ettore Biondi [1] & Zhongwen Zhan[1]

Earthquake focal mechanisms provide critical in-situ insights about the sub-surface faulting geometry and stress state. For frequent small earthquakes (magnitude< 3.5), their focal mechanisms are routinely determined using first-arrival polarities picked on the vertical component of seismometers. Nevertheless, their quality is usually limited by the azimuthal coverage of the local seismic network. The emerging distributed acoustic sensing (DAS) technology, which can convert pre-existing telecommunication cables into arrays of strain/strain-rate meters, can potentially fill the azimuthal gap and enhance constraints on the nodal plane orientation through its long sensing range and dense spatial sampling. However, determining first-arrival polarities on DAS is challenging due to its single-component sensing and low signal-to-noise ratio for direct body waves. Here, we present a data-driven method that measures P-wave polarities on a DAS array based on cross-correlations between earthquake pairs. We validate the inferred polarities using the regional network catalog on two DAS arrays, deployed in California and each comprising ~ 5000 channels. We demonstrate that a joint focal mechanism inversion combining conventional and DAS polarity picks improves the accuracy and reduces the uncertainty in the focal plane orientation. Our results highlight the significant potential of integrating DAS with conventional networks for investigating high-resolution earthquake source mechanisms.

Source mechanisms of seismic events are essential for characterizing the faulting process, rupture plane geometry, fluid-solid interaction, and in-situ medium properties[1-4]. For tectonic crustal earthquakes, analyzing stress state and variation derived from a group of source mechanisms improves our understanding of earthquake-stress interaction[5,6] and contributes to the evaluation of seismic hazards[7,8]. Source mechanisms of larger earthquakes (magnitude≥3.5) are usually determined by fitting observed waveforms with synthetics[9-11]. For the more frequent smaller earthquakes (magnitude<3.5), their focal mechanisms (i.e. the fault plane solution) generally rely on the first-arrival P-wave polarities[12-15], which can be determined through manual picking, bayesian approach, or deep learning[16-18]. Although these focal mechanisms can be further constrained by including P- and S-wave amplitude ratios[19,20], single-event-based focal mechanism determination of small earthquakes tends to have low quality due to the lack of

reliable polarity picking. Recently, by taking account of the relative polarities and amplitude ratios between earthquake pairs, a 'composite' focal mechanism for a cluster of earthquakes can be constrained with higher accuracy and applied to even smaller earthquakes[21-25]. Such methods require a well-established template database with known polarity picks for historical earthquakes. The focal mechanism accuracy also depends on the coverage of the local seismic network and the spatial distribution of seismicity.

Distributed acoustic sensing (DAS) technology has emerged as a revolutionary seismic monitoring tool, capable of converting fiber-optic cables into dense seismic arrays that extend up to 100 kilometers[26-28]. By sending laser pulses and measuring the phase change of Rayleigh back-scattering from intrinsic fiber-cable impurities, a DAS interrogator unit measures the longitudinal strain or strain rate along either dedicated fiber cable or pre-existing

---

[1] Seismological Laboratory, Division of Geological and Planetary Sciences, California Institute of Technology, Pasadena, CA, USA. ✉e-mail: jxli@caltech.edu

telecommunication fiber cables[27,28]. DAS' wide dynamic range at a broad frequency band has enabled seismic monitoring of earthquakes, volcanic events, and glacial-related seismicities[29–35]. These studies mainly focus on seismic detection, location, and magnitude estimation utilizing spectral, traveltime, and amplitude information. However, the determination of first-arrival polarities - critical for investigating source mechanisms - has been missing due to DAS' single-component sensing and weak sensitivity to the direct P-wave arrivals.

Several studies used borehole DAS arrays to characterize source mechanisms of microseismic events by comparing theoretical predictions and synthetic modelings with observed waveforms[36–39]. Additionally, a case study in Antarctica performed waveform-based inversion of the icequake source mechanism using a one-kilometer-long fiber buried in the snow[40]. While these studies in special settings demonstrate the potential of using DAS for studying source mechanisms, challenges remain in leveraging the surface fiber cables (such as the extensive pre-existing telecommunication fiber cables) for such purposes. DAS recordings along these fiber cables usually experience higher ambient noise levels, stronger surface scatterings, complex cable geometries, and unknown coupling conditions compared to microseismic monitoring using dedicated fibers cemented in the borehole or snow. Waveform-based source mechanism inversion may be hindered by inaccurate Green's functions due to unaccounted structural or coupling complexities[41]. Additionally, previous polarity determination methods such as deep learning or relative measurement face challenges due to the lack of training labels or well-established template databases on DAS recordings[17,18,21].

In this study, we introduce a data-driven approach that enables the determination of P-wave polarities along a DAS array through cross-correlations between earthquake pairs. Our method differs from standard relative measurements, which rely on known polarities from template events to determine unknown polarities of new events. Instead, our relative measurements are all among events with unknown polarities. We take advantage of DAS' dense spatial sampling and waveform similarities between adjacent channels to obtain the polarities that are consistent across all recording channels. We apply this method to two DAS arrays and validate the inverted polarities across ~5000 channels by comparing them with the predicted polarities derived from the catalog focal mechanism. We find that inverted polarities correspond directly to the first-arrival vertical displacement polarity rather than the longitudinal strain polarity. The inferred polarity reversals along the DAS array contribute to the accurate determination of focal plane orientation. By conducting a joint focal mechanism inversion using both conventional and DAS P-wave polarity picks, we are able to systematically improve the focal mechanism quality for every single earthquake.

## Results

### Measure polarities using a relative approach

We recorded data using two DAS arrays in California, one located in Long Valley Caldera, and the other in Ridgecrest. These two DAS arrays have similar interrogation setups, each contains 5000 channels with a 10 meter channel spacing and a total recording length of approximately 50 kilometers. Both datasets have been resampled to 100 Hz. Although our DAS arrays can record clear P-wave and S-wave onset of local small earthquakes (Fig. 1a), picking the P-wave polarity directly from the strain-rate waveform is challenging for two reasons: First, the DAS recording has lower signal-to-noise ratios compared to conventional broadband stations. Second, DAS recording is more sensitive to locally scattered surface waves than vertical particle motions induced by incident direct P waves because the DAS unit is interrogating along a horizontal fiber. Nevertheless, the channel-by-channel cross-correlations between the P-wave windows of earthquake pairs can

exhibit clear correlation peaks (Fig. 1d–f) as the patterns of local scatterers are similar for different earthquakes. A positive correlation coefficient indicates that those two earthquakes share the same polarity, while a negative correlation value implies an opposite polarity on a given channel. The clear correlation peaks and sharp relative-polarity reversals (with a resolution of tens of meters) across channels show that these cross-correlations contain robust high-spatial-resolution relative polarity information.

Based on these observations, we devise a method to derive P-wave polarities from cross-correlations among a cluster of earthquakes that share similar waveforms in the P-wave window (see Methods). In a standard relative polarity measurement, treating each channel independently allows us to infer the unknown polarities from known ones of template events through the singular value decomposition (SVD) of the relative polarity matrix[21] (see Methods). However, since all our measurements are among earthquakes with unknown P-wave polarities, there exists a sign ambiguity at each channel (see Methods). Thanks to the ultra-dense spatial sampling of the DAS array, the waveforms in the P-wave window are similar on neighboring channels. By incorporating the cross-correlations between adjacent channels for all earthquake pairs, we can resolve the sign ambiguity between channels, resulting in corrected polarities that are consistent across all channels. We refer to this as "channel-consistent" polarity. Since all aforementioned measurements are performed in a relative sense, the obtained polarity still contains one sign ambiguity with respect to the absolute polarity of all earthquakes and all channels. This ambiguity can in principle be corrected by determining one polarity pick on a conventional sensor close to the DAS array. In practice, the ambiguity correction is more robust using polarities of multiple events on one or more nearby conventional sensors.

### Inverted polarities as the P-wave vertical displacement polarity

We apply our approach to the two DAS arrays. For the Long Valley Caldera DAS array, we test the method on 25 local earthquakes that share similar waveforms and have magnitudes ranging from 1 to 3.4. We extract the P-phase window using a deep-learning phase picker[42], perform cross-correlations for all the earthquake pairs, and determine the correlation peaks using multi-channel cross-correlations (MCCC) (see the workflow in Fig. S1)[43]. With about 3 million relative polarity picks, we invert the "channel-consistent" polarities for the 25 earthquakes across ~ 5000 channels. Using the focal mechanism catalog from the Northern California Earthquake Data Center (NCEDC)[44], we can predict the observed polarity on all DAS channels. We find that the inverted polarities match well with the predicted vertical displacement polarities instead of the predicted longitudinal-strain polarities (Fig. 2).

Similarly, for the Ridgecrest DAS array, we applied the same procedure to 30 local earthquakes. The inverted polarities demonstrate good agreement with the predicted vertical displacement polarities using the focal mechanisms from the Southern California Earthquake Data Center (SCEDC)[45]. The underlying physical mechanism for these results is that the primary contributions to the cross-correlations between P-wave windows of earthquake pairs come from surface scatterings since a horizontal fiber cable has weaker sensitivity to the near-vertical first motion. Moreover, the sign of the surface scatterings directly corresponds to the direction of initial movement caused by incident P waves. Therefore, even though it is hard to directly determine first-motion polarity from the raw waveforms due to DAS' low signal-to-noise ratio and horizontal sensitivity, we can robustly infer the relative first-motion polarity between earthquake pairs. More specifically, a positive cross-correlation value between two P-wave windows indicates the same initial movement direction of incident P waves, while a negative value indicates an opposite initial movement direction.

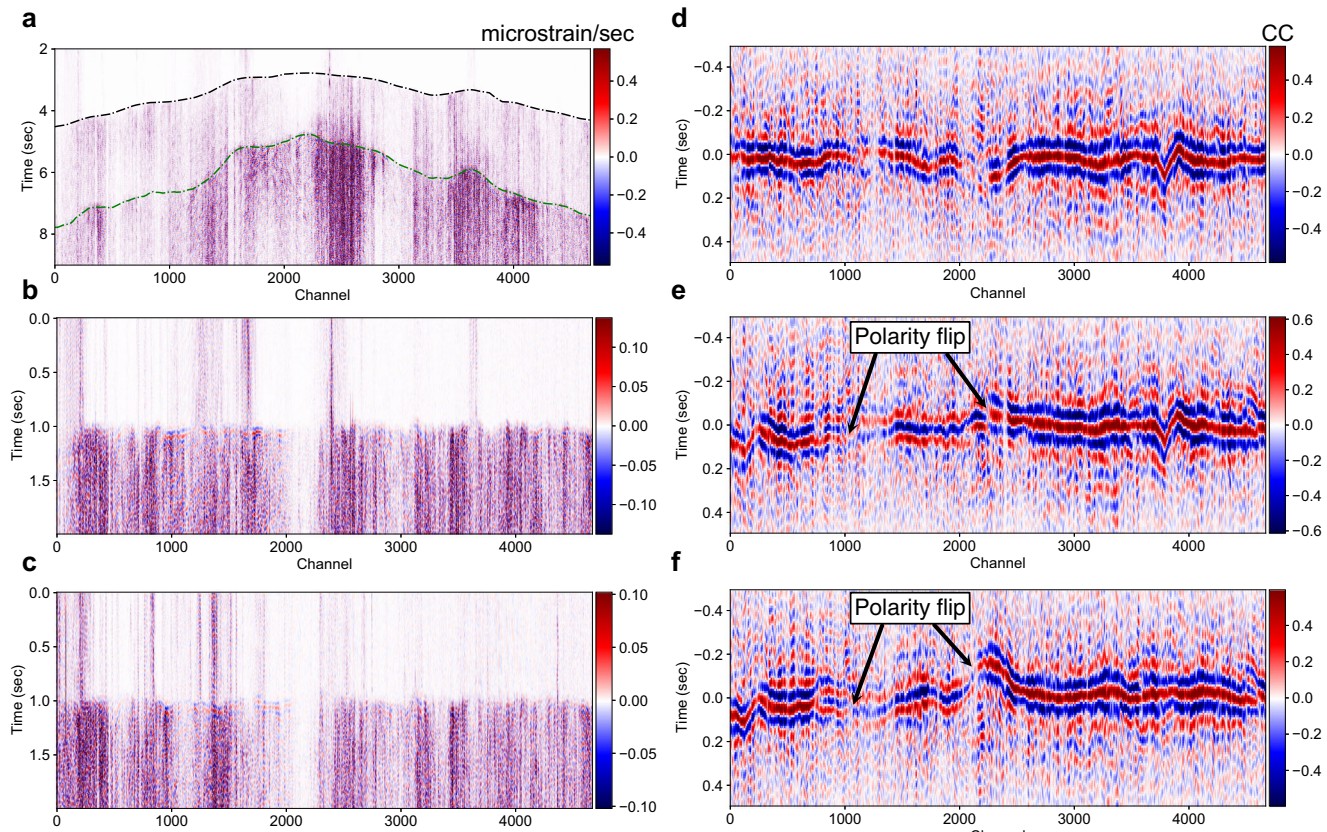

**Fig. 1 | Retrieval of relative polarity information through cross-correlations between P-wave windows of earthquake pairs. a**. Example microstrain rate recording of a local M2.5 earthquake band-pass filtered between 1 to 10 Hz. The DAS recording shows clear P and S wave onset marked by the black and green dashed lines. **b**. P-wave window extracted along the black-dashed line. The strong scattering and low signal-to-noise ratio make polarity picking on raw DAS waveforms challenging. **c**. P-wave window extracted from another M3.4 earthquake. **d**. Channel-by-channel cross-correlation between P-wave windows in **b** and **c**, with a 10-channel moving average applied along the channel axis to enhance the signal-to-noise ratio. Clear positive cross-correlation peaks (red color) indicate similar P-wave polarities between two earthquakes across all channels. **e** Channel-by-channel cross-correlation between **b** and the P-wave window of another M2.8 earthquake. There are clear polarity flips at channels around 1000 and 2000. The negative cross-correlation peaks (blue color) in the middle indicate that these two earthquakes share different P-wave polarities on those channels. **f** Similar to **e**, but the cross-correlation between **c** and the same M2.8 earthquake.

## Improved focal mechanisms with DAS polarity picks

After the verification of our method on two small clusters of earthquakes, we extend the analysis to a larger number of earthquakes using the Long Valley Caldera DAS array. Specifically, we perform relative polarity measurements for 147 local earthquakes with magnitudes ranging from 0.5 to 3.4 (Fig. 3a). Following the same workflow, we derive the polarities for all 147 earthquakes across all recording DAS channels. We then compare the inverted polarities with the ones predicted by the focal mechanisms from the NCEDC catalog (Fig. 3b, c). Even though some earthquakes have no focal mechanisms available from the catalog (white vertical columns in Fig. 3c), we can still robustly invert for their polarities along the DAS array. In general, the inverted polarities show good agreement with the predicted ones. However, the polarity-reversal points predicted from the catalog appear to be more scattered compared to our inverted results. This scattering results from the errors in the catalog focal mechanism.

The NCEDC catalog determines the focal mechanism based on first-arrival polarities[12], and their accuracy is usually limited by the azimuthal coverage. With the inverted DAS polarities, we can fill the azimuthal gap by performing a joint focal mechanism inversion using both conventional and DAS polarity picks. In addition to the improvement of the azimuthal coverage, the inverted polarity flips along the fiber cable can tightly constrain the focal plane orientation. In Fig. 4, we show two examples of joint focal mechanism inversion.

With the inverted polarity flips, all acceptable focal planes must intersect the transition point and separate the positive and negative segments of polarities, which can improve the accuracy and lower the uncertainty. With multiple inverted polarity flips (Fig. 4d), we can uniquely determine the focal mechanism, with uncertainties nearly contributed solely by the ray paths. Notably, for the example in Fig. 4d, a focal mechanism inversion using only DAS polarity picks can achieve the same resolution (Fig. 4e). This example demonstrates that one DAS array is sufficient to fully constrain the focal mechanism given enough sampling in different quadrants. More strictly speaking, if the DAS can sample at least once across one nodal line and twice across the other nodal line, the focal mechanism (the fault plane) solution can be uniquely determined. Overall, by incorporating the inverted DAS polarities, we can systematically improve the focal mechanism quality (see definition in Methods) for individual earthquakes (Fig. 3). More specifically, the root-mean-square (RMS) angle differences of the accepted solutions from the preferred solution have been decreased by 15° on average.

Lastly, we further validate the inverted DAS polarities and the improvement of jointly inverted focal mechanisms by examining a cluster of linearly distributed earthquakes. The northeast-southwest linear trend of this earthquake cluster suggests that these earthquakes may be located on the same geological faults and share similar focal plane orientations (inset figure in Fig. 3a). Indeed, the inverted DAS polarities of those earthquakes show considerable similarities, while

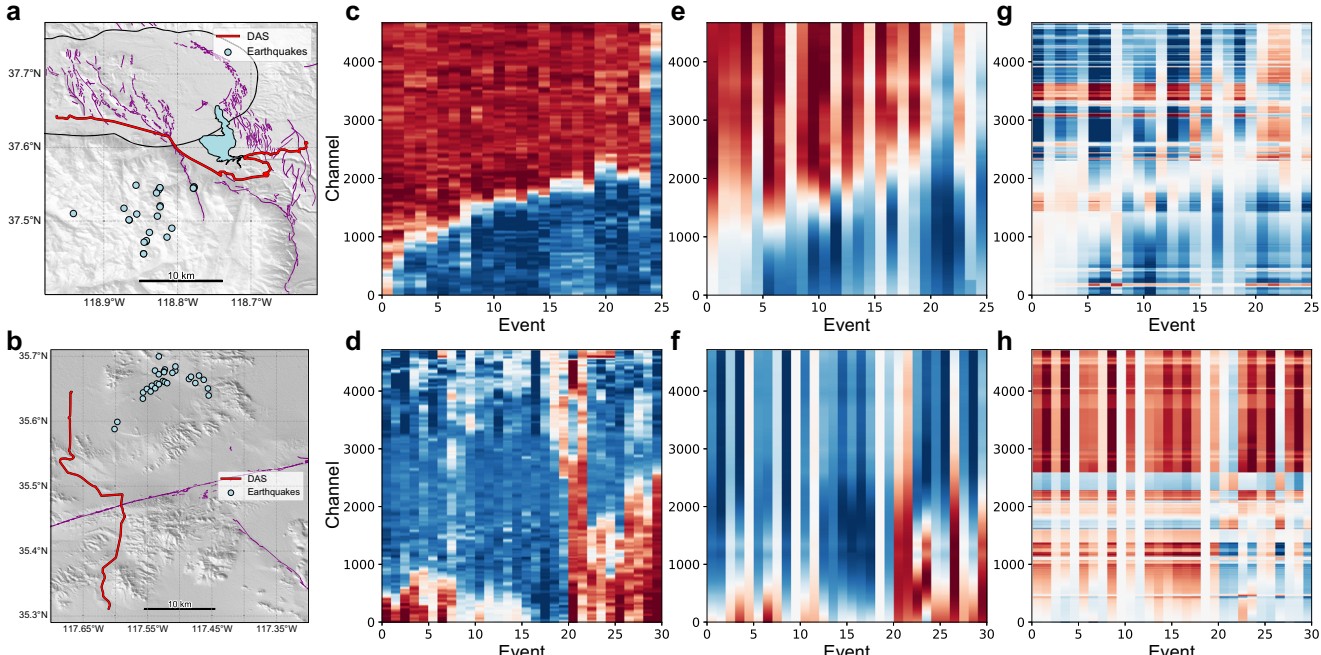

**Fig. 2 | Inverted DAS polarities as P-wave vertical displacement polarities verified on two DAS arrays.** We verify our method using two DAS arrays, one near the Long Valley Caldera, CA (**a**, **c**, **e**, **g**), and the other near Ridgecrest, CA (**b**, **d**, **f**, **h**). Each DAS array has about 5000 channels and a 10-meter channel spacing. **a** 25 testing earthquakes (light blue dots) to the south of the DAS array (red line) near the Long Valley Caldera (black closed circle) hosting many faults (solid purple lines)[46]. The background gray color indicates the topography[47]. **b** Similar to **a**, 30 testing earthquakes to the northeast of the Ridgecrest DAS array. **c**, **d** Inverted polarities of testing earthquakes on two DAS arrays using relative polarity measurements. **e**, **f** Predicted vertical displacement polarities on two DAS arrays using catalog focal mechanisms. **g**, **h** Predicted longitudinal strain polarities on two DAS arrays using catalog focal mechanisms. The inverted DAS polarities (**c**, **d**) are consistent with the predicted vertical displacement polarities (**g**, **h**) instead of longitudinal strain polarities (**e**, **f**) from catalog focal mechanisms.

the predicted polarities using catalog focal mechanisms show greater variations among events (Fig. 3b, c). The jointly inverted focal mechanisms show more concentrated compression (P) & tension (T) axes compared to the catalog focal mechanisms (Fig. 3e, f). Moreover, the inverted strike angles align well with the northeast-southwest linear trend of the earthquake distribution (inset figure in Fig. 3a).

## Discussion

The rapidly developing DAS technology is converting extensive onshore or offshore fiber cables into sensitive seismic antennas. For the fast-growing DAS dataset, previous studies with DAS mainly explore seismic detections and locations. While the source mechanism is an essential seismic attribute, there have been only a few attempts using DAS through comparing or fitting synthetic waveforms with observed ones. These methods, however, suffer from inaccurate Green's functions for recordings on surface fiber cables due to unknown site couplings and strong surface scatterings. In contrast, our proposed method provides a data-driven approach that can densely sample polarity variations over tens of kilometers. By integrating DAS polarities into the current seismic network, we can systematically improve the focal mechanism inversion quality.

The experiments with two onshore DAS arrays highlight the unique advantages of DAS over conventional seismic sensors, i.e. the long sensing range (tens of kilometers) and high spatial sampling (tens of meters). We can obtain high-resolution focal mechanisms if the focal plane orientation intersects with the recording fiber cable. Theoretically, if the fiber cable samples across one nodal line once and the other nodal line twice on the beach ball, the focal mechanism can be uniquely determined, since the focal mechanism has only three independent parameters. The remaining uncertainties will be attributed to takeoff angles due to inaccurate earthquake locations and unaccounted velocity structures. These

results motivate the design of optimal fiber geometries for monitoring microseismicity in fields such as the geothermal reservoir or carbon storage and sequestration (CSS) site. Our method is also applicable to offshore fiber cables, where the deployment of ocean-bottom seismometers is sparse and costly[27].

Currently, our investigation focuses on determining P-wave polarities using DAS. By performing relative S-wave polarity and P/S amplitude ratio measurements, we can derive additional constraints for studying focal mechanisms with DAS. Furthermore, beyond the fault plane solution, DAS can potentially facilitate the characterization of the non-double-couple components, which is critical to revealing the complex underlying rupture mechanisms. More specifically, we can use DAS polarities to cross-validate the full moment tensor solution derived by conventional stations. However, it is important to note the limitations of our method. There is a high computation cost due to the large number of DAS channels. For example, performing MCCC on thousands of DAS channels would require solving a large sparse matrix through least-squares (see Methods). Such problems can be addressed through the implementation of a GPU-based MCCC picker or by training a deep-learning-based MCCC picker. Lastly, a numerical study should be conducted to systematically investigate the response of correlation functions to surface scatterings caused by topography or velocity heterogeneities.

Note that, once polarities are obtained on one DAS array for a cluster of earthquakes, their P-wave windows can act as known templates or labels. We can then use them to determine P-wave polarities of new events through either relative measurements or deep learning[17,18,21]. For example, by cross-correlating the continuous recordings using the established DAS template database, we can detect smaller earthquakes using the conventional template-matching technique. Meanwhile, if clear correlation peaks exist across channels, we can use the MCCC to pick DAS-based differential traveltime and

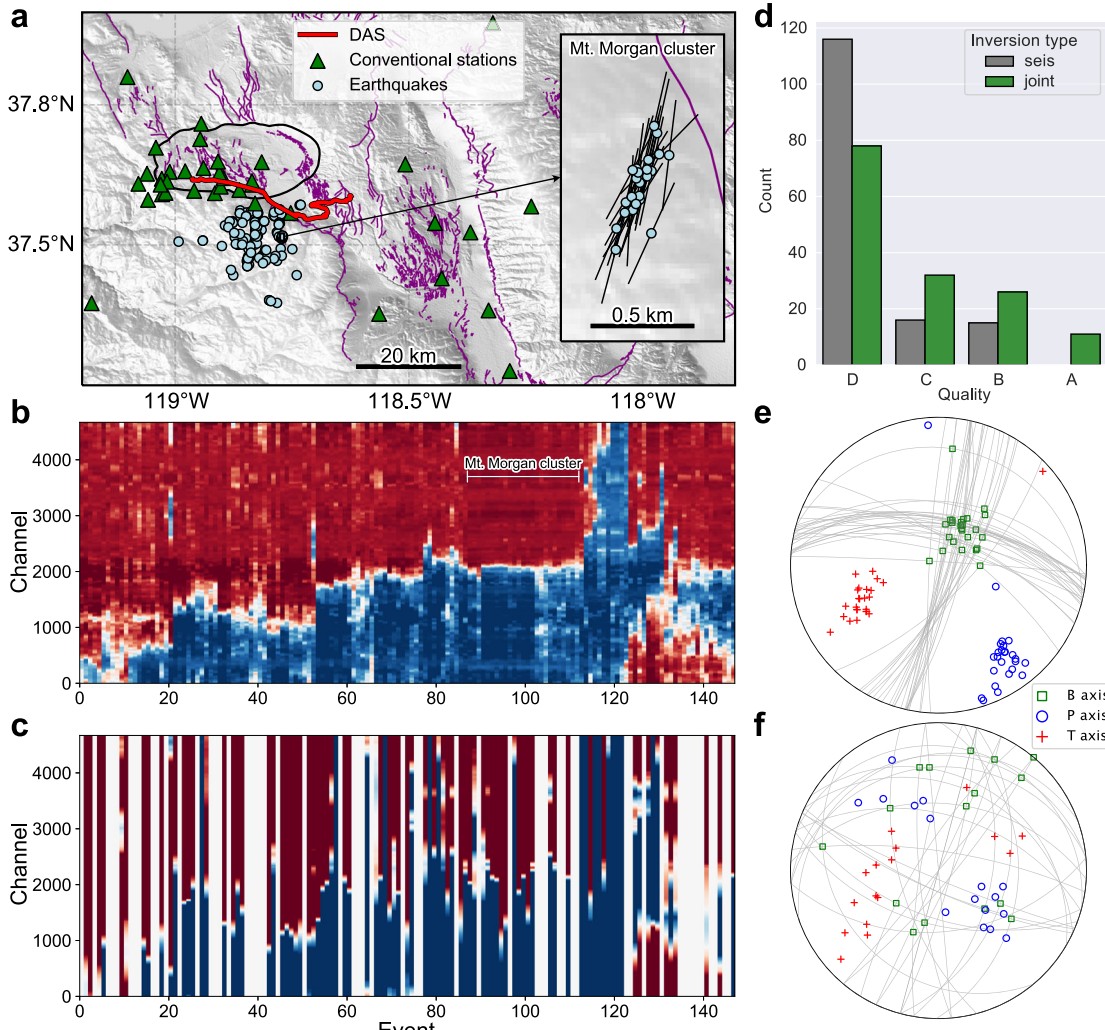

**Fig. 3 | Inverting DAS polarities for more earthquakes near the Long Valley Caldera and improving focal mechanism quality through joint inversion.**
**a** Similar to Fig. 2a, here shows the map view of a larger region near the Long Valley Caldera with earthquake locations (147 earthquakes marked as light blue dots), fiber cable geometry (red line), and local seismic network (green triangles) indicated. The zoom-in map shows a cluster of 25 earthquakes exhibiting a clear northeast-southwest linear trend near Mt. Morgan, suggesting that these earthquakes may be located on the same fault and share similar focal mechanisms. The black lines represent strike orientations of jointly inverted focal mechanisms.
**b** Inverted DAS polarities along the fiber cable for all 147 earthquakes. The inverted

polarities for the Mt. Morgan cluster are indicated by the horizontal white bar.
**c** Predicted P-wave vertical displacement polarities using catalog focal mechanisms. White columns represent earthquakes without a focal mechanism solution in the NCEDC catalog[44]. **d**. Comparison of obtained focal mechanism quality between inversion using only conventional polarity picks (seis) and joint inversion (joint) using both conventional and DAS polarity picks. **e** Focal mechanism nodal lines (gray lines) and TBP axes of jointly inverted focal mechanisms. Here, T represents the tension axis, P represents the pressure axis, and B is defined as the right-hand vector product between T-axis and P-axis. **f** Focal mechanism nodal lines and TBP axes of catalog focal mechanisms.

relative polarity for high-resolution earthquake relocation and source mechanism inversion. By routinely streaming the data, we anticipate DAS to continuously produce robust polarity picks for pursuing a better focal mechanism catalog and a more accurate understanding of tectonic stress distributions and variations.

## Methods

### Workflow

To obtain jointly inverted focal mechanisms from the continuous DAS recordings, our workflow mainly consists of three steps (Fig. S1): pre-processing, polarity picking & inversion, and joint focal mechanism inversion. In the pre-processing step, we use PhaseNet-DAS to extract a 2-second P-phase window from the continuous data[42]. The waveform is band-pass filtered at a frequency band of 1 to 10 Hz, and a median filter is applied to eliminate the common mode noise[31]. During the second step, we perform pairwise cross-correlations among all similar P-phase windows at each channel and its adjacent channel. We use the MCCC to

pick the maximum absolute cross-correlation values, where the sign of the picked value indicates the relative polarity. The picked relative polarities are then inverted to obtain the "channel-consistent" polarities for all earthquakes and all channels. The final absolute polarity can be obtained by measuring one or more robustly, multiple P-wave polarities on the vertical components of nearby broadband stations. In the final step, we apply a first-arrival polarity-based joint focal mechanism inversion using both conventional polarity picks and inverted DAS polarity picks. The conventional polarity picks are downloaded from the NCEDC[44].

### Relative polarity measurement through iterative multi-channel cross-correlation

By picking the maximum absolute cross-correlation values on the P-wave correlograms, we can derive two important pieces of information: The first one is the relative polarity, which can be represented by the sign of maximum absolute cross-correlation. The

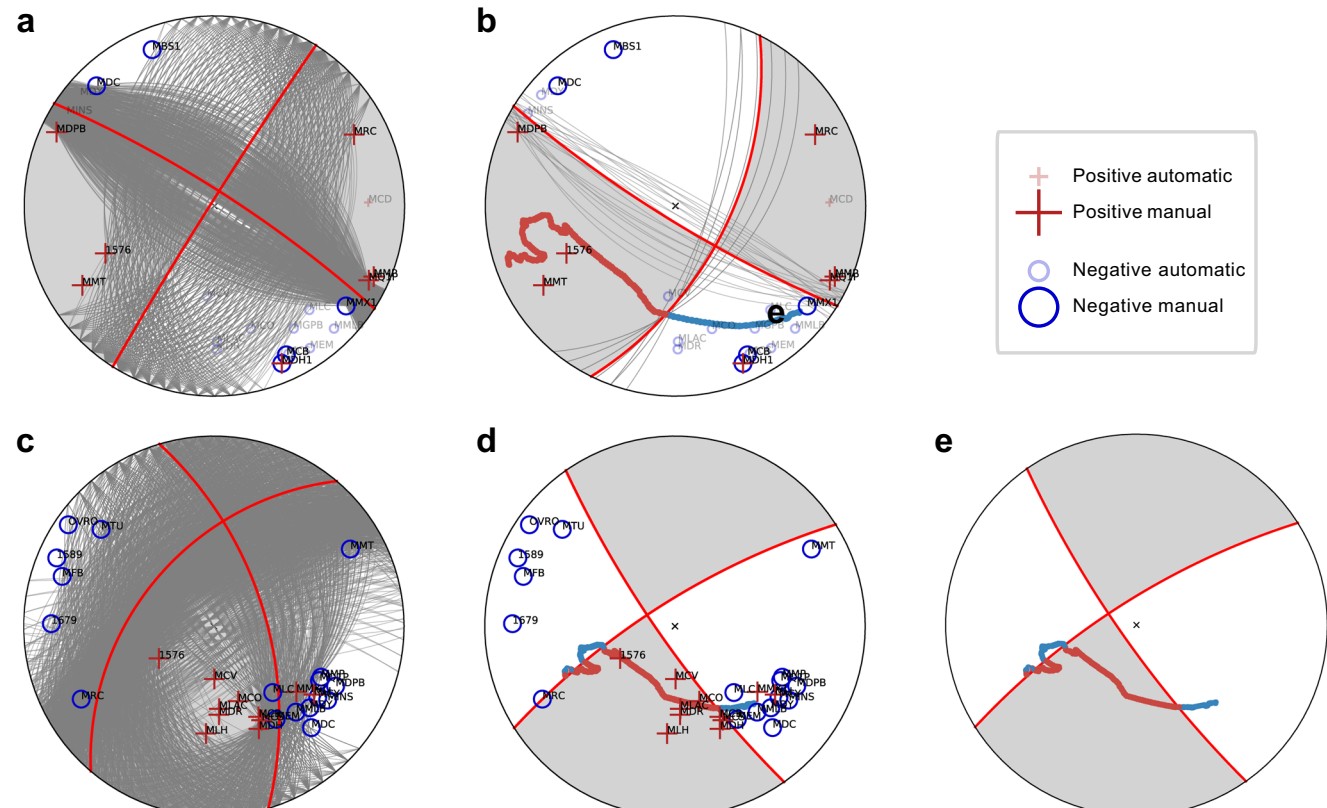

**Fig. 4 | Improving the accuracy and reducing the uncertainty in the focal plane orientation with DAS polarities. a**, **b** Average focal mechanism solution (red line, with gray background color representing positive first motion) and all accepted solutions (gray lines) for an M1.7 earthquake (event ID: NC73566395). The focal mechanism solution in **a** uses only conventional polarity picks (crosses and circles with their respective station names indicated). The focal mechanism solution in **b** uses both conventional and DAS polarity picks (continuous red and blue dots).

The polarity changes from red to blue along the fiber cable require the nodal lines to intersect the polarity-flip points. **c**, **d** Focal mechanism solutions for another M2.8 earthquake (event ID: NC73482516). The focal mechanism solution using only conventional picks in **c** indicates a reverse-faulting mechanism, while the jointly inverted focal mechanism in **d** reveals a strike-slip mechanism. **e** The focal mechanism solution can also be uniquely determined using only DAS polarities if the fiber cable samples across the nodal lines multiple times.

second one is the differential delayed time at each channel, which quantitatively measures the deep-learning picking error. Since the DAS array samples the wavefields densely and continuously in space, the correlograms at nearby channels look similar and are varying smoothly along the channel axis. By performing cross-correlations between neighboring correlograms (maximum interval of 10 channels in our case), and measuring the double-differential delay times of the two-fold correlation peaks, we form an over-determined linear system (equation (1)) and solve the double-differential delayed times using least-squares. This approach is known as the multi-channel cross-correlation as described by Van-Decar and Crosson[43].

$$\mathbf{D}\boldsymbol{\tau} = \Delta\boldsymbol{\tau}, \tag{1}$$

where $\mathbf{D}$ is the differential operator, $\boldsymbol{\tau}$ is the absolute differential delayed time of the correlation peak on the correlogram, and $\Delta\boldsymbol{\tau}$ is the measured double-differential delayed time.

The solution to equation (1) is not unique and can be added with any constants. To determine the unique delayed time and pick the signed cross-correlation peaks (relative polarities), we pick the delayed time using the global maximum absolute of the correlograms on each channel and incorporate them in the linear system (equation (2)). We then iteratively update the pick in a shrinking picking window centered

around the pick until the picking window is less than 0.05 s.

$$\begin{bmatrix} \lambda\mathbf{D} \\ \mathbf{I} \end{bmatrix} \boldsymbol{\tau} = \begin{bmatrix} \lambda\Delta\boldsymbol{\tau} \\ \boldsymbol{\tau}_p \end{bmatrix}, \tag{2}$$

where $\mathbf{I}$ is an identity matrix, $\boldsymbol{\tau}_p$ is the picked delayed time, and $\lambda$ is the weight for double-differential smoothing, which is set to 1 in this study.

### Inverting channel-consistent DAS polarities from relative measurements

Consider a cluster of $N$ similar earthquakes observed at $K$ stations. For each event $i$, there is an unknown polarity $P_{ik}$ recorded at station $k$, where $i = 1, \cdots, N$, and $k = 1, \cdots, K$. The unknown polarity vector $\overrightarrow{\mathbf{p}}_k = [P_{1k}, P_{2k}, \ldots, P_{Nk}]^T$ represent polarities recorded at station $k$ for all the N events. Directly determining the polarity is challenging due to the low signal-to-noise ratio and unknown site factors for small earthquakes. However, since these site factors at the same station are similar for different earthquakes, the signs of their cross-correlations or relative polarities are easier to determine. In the case of conventional relative measurements[21], we have $M$ template events with known polarities: $\overrightarrow{\mathbf{q}}_k = [Q_{1k}, Q_{2k}, \ldots, Q_{Mk}]^T$. We can then form a relative measurement matrix, which is ideally a rank-one matrix formed by the outer product of unknown polarity vector $\overrightarrow{\mathbf{p}}_k$ and template polarity

vector $\overrightarrow{\mathbf{q}}_k$ at each station $k$:

$$\mathbf{R}_{NM}^{<kk>} = \overrightarrow{\mathbf{p}}_k \otimes \overrightarrow{\mathbf{q}}_k = \begin{bmatrix} P_{1k}Q_{1k} & P_{1k}Q_{2k} & \cdots & P_{1k}Q_{Mk} \\ P_{2k}Q_{1k} & P_{2k}Q_{2k} & \cdots & P_{2k}Q_{Mk} \\ \vdots & \vdots & & \vdots \\ P_{Nk}Q_{1k} & P_{Nk}Q_{2k} & \cdots & P_{Nk}Q_{Mk} \end{bmatrix}. \quad (3)$$

Here, the superscript $^{<kk>}$ denotes that the relative measurement is performed at the same station $k$. The subscript $_{NM}$ denotes that the relative measurement is between $N$ unknown events against $M$ template events. The $\otimes$ denotes the outer product operation between two vectors. We can perform a singular value decomposition (SVD) to the above rank-one matrix:

$$\mathbf{R}_{NM}^{<kk>} = \mathbf{U}_{NM}^{<kk>} \mathbf{S}_{NM}^{<kk>} \mathbf{V}_{NM}^{<kk>*}, \quad (4)$$

where the $\mathbf{U}_{NM}^{<kk>}$ and $\mathbf{V}_{NM}^{<kk>*}$ are orthogonal matrices and $\mathbf{S}_{NM}^{<kk>}$ is a diagonal matrix with singular values sorted in descending order. The first column vector of matrix $\mathbf{U}_{NM}^{<kk>}$ will then correspond to the unknown polarity vector $\overrightarrow{\mathbf{p}}_k$[21].

However, for the DAS measurements across thousands of channels, we do not have template events with known polarities. The relative measurement matrix $\mathbf{R}_{NN}^{<kk>}$ will be all among events with unknown polarities and is a symmetric matrix formed by the outer product of $\overrightarrow{\mathbf{p}}_k$ itself:

$$\mathbf{R}_{NN}^{<kk>} = \overrightarrow{\mathbf{p}}_k \otimes \overrightarrow{\mathbf{p}}_k = \mathbf{U}_{NN}^{<kk>} \mathbf{S}_{NN}^{<kk>} \mathbf{U}_{NN}^{<kk>*} \quad (5)$$

The first column vector $\overrightarrow{\mathbf{u}}_k^{<kk>}$ of matrix $\mathbf{U}_{NN}^{<kk>}$ from the SVD of $\mathbf{R}_{NN}^{<kk>}$ will correspond to the unknown polarity vector $\overrightarrow{\mathbf{p}}_k$. However, there is a sign ambiguity at each channel $k$ since the SVD still holds if we multiply $\mathbf{U}_{NN}^{<kk>}$ with a "-1". This sign ambiguity at $K$ channels will result in $2^K$ different possible combinations of inverted polarity vectors.

To resolve this ambiguity, we leverage the high spatial sampling of the DAS array. Since the DAS recording channel spacing (10 meters) is much smaller than the wavelength (> hundreds of meters), the waveforms recorded at nearby channels are similar. The cross-correlations at different channels $k$ and $l$ can still produce high cross-correlation values and robust relative measurements when $|k - l|$ is small. In particular, we further calculate pairwise cross-correlations for neighboring channels ($k = l - 1$) and use the MCCC technique to pick the relative polarities and form the relative measurement matrix $\mathbf{R}_{NN}^{<kl>}$:

$$\mathbf{R}_{NN}^{<kl>} = \overrightarrow{\mathbf{p}}_k \otimes \overrightarrow{\mathbf{p}}_l = \mathbf{U}_{NN}^{<kl>} \mathbf{S}_{NN}^{<kl>} \mathbf{V}_{NN}^{<kl>*} \quad (6)$$

Similarly, the first column vector $\overrightarrow{\mathbf{u}}_k^{<kl>}$ of $\mathbf{U}_{NN}^{<kl>}$ matrix and the first row vector $\overrightarrow{\mathbf{v}}_k^{<kl>}$ of $\mathbf{V}_{NN}^{<kl>}$ from the SVD of $\mathbf{R}_{NN}^{<kl>}$ correspond to the unknown polarity vector $\overrightarrow{\mathbf{p}}_k$ and $\overrightarrow{\mathbf{p}}_l$. Although the sign ambiguity still exists, the sign of their multiplication $\overrightarrow{\mathbf{u}}_k^{<kl>} \cdot \overrightarrow{\mathbf{v}}_l^{<kl>}$ is unique because the two "-1"s cancel out in the multiplication. The sign must also be the same as the sign of the multiplication between $\overrightarrow{\mathbf{u}}_k^{<kk>} \cdot \overrightarrow{\mathbf{u}}_l^{<ll>}$ through the connection of ground-truth relative polarity $sign(\overrightarrow{\mathbf{p}}_k \cdot \overrightarrow{\mathbf{p}}_l)$. We then use the following relationship to correct the sign ambiguity of SVD in equation (5):

$$sign\left(\overrightarrow{\mathbf{u}}_k^{<kk>} \cdot \overrightarrow{\mathbf{u}}_l^{<ll>}\right) = sign(\overrightarrow{\mathbf{p}}_k \cdot \overrightarrow{\mathbf{p}}_l) = sign\left(\overrightarrow{\mathbf{u}}_k^{<kl>} \cdot \overrightarrow{\mathbf{v}}_l^{<kl>}\right) \quad (7)$$

After the correction using cross-correlations on neighboring channels, we can still multiply all the inverted polarities by "-1" simultaneously, and the equations (5) and (6) still hold. This behavior is because all measurements are performed in a relative sense. We can correct this last sign ambiguity by picking one P-wave polarity for one of the earthquakes, or more robustly, multiple P-wave polarities from multiple earthquakes recorded at collocated conventional seismometers.

## Joint focal mechanism inversion

We perform a grid search for the focal mechanism using both conventional and DAS P-phase polarity picks. The conventional polarity picks are downloaded from the NCEDC[44]. The objective function is the ratio of mispredicted polarities averaged between conventional and DAS polarity picks. The acceptable solutions satisfy misfit ratio thresholds independently for conventional and DAS polarity picks. The misfit ratio threshold, defined as the ratio of inconsistent polarity predictions to the total number of polarity picks[13], is set to 15% for conventional polarity picks and 1% for DAS polarity picks through trial and error. In future applications of this method, these parameters can be adjusted as hyperparameters based on the quality of conventional and DAS polarity picks, and the consistency of the joint solution between them. For example, we can perform a grid search on these hyperparameters for specific DAS cables in different regions. We then determine the focal mechanism solution using the average of all accepted solutions. The quality of the solution is defined in the same way as described in HASH[13].

## Data availability

The catalog focal mechanisms and conventional polarity picks are available from the NCEDC and SCEDC data centers. The DAS P-phase dataset used in this study is available at https://doi.org/10.22002/n47vy-s0s65.

## Code availability

The codes are available upon request to the authors.

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

## Acknowledgements

This work is supported by the US National Science Foundation (NSF) (EAR-1848166, Z.Z.) and the Gordon and Betty Moore Foundation (Z.Z.).

## Author contributions

J.L. conceived the idea, implemented the code, and conducted the experiment. W.Z. performed the phase picking. J.L. and E.B. developed the method. Z.Z. advised the project. All authors contributed to the writing and reviewing of the manuscript.

## Competing interests

The authors declare no competing interests.
