## [Peer Review File · Nature Communications]

Earthquake focal mechanisms with distributed acoustic sensingREVIEWERS' COMMENTS

Reviewer #1 (Remarks to the Author):

I really enjoyed reading this manuscript. It was clear and concise, and represents a step change in a fundamental earthquake parameter estimation technique – the determination of the orientation of the fault motion during an earthquake. The subject of the manuscript centers on the use of exciting new technology – the use of optical fiber networks as seismometer arrays – and develops an intricate data processing strategy to leverage this new form of data to improve the field.

There have been many journal articles about fiber-optic Distributed Acoustic Sensing (DAS) technology written in the recent years, and a few high impact articles in Nature Communications (e.g., Jousset et al., 2018; Williams et al., 2019; Munoz and Soto, 2022). The topic is high impact and publicly interesting because of how it has the potential to dramatically improve the field of earthquake science and geophysics, in general. However, many articles have only advertised “potential” impacts – this manuscript actually documents an impact. As such, I think it will be high impact among earth scientists.

I have a few comments and questions for the authors, but my recommendation is to accept the manuscript with minor revisions.

Basic typos:

P3, last sentence – suggest “scatters” should be “scatterers”

Figure 2 caption – suggest “b. Earthquake location” should be “b. Earthquake locations”

More significant comments:

Figure 4 is very powerful because it highlights the improvement in focal mechanism estimation enabled by DAS. Still, I was a bit confused at first because the focal plane solution in b for NC73566395 is not an acceptable solution in a. As discussed in Methods the acceptable misfit ratio is 15% for inertial sensor network and 1% for DAS. For the inertial sensor network only, what is the misfit ratio a ratio of? How were the numbers 15% and 1% selected? Bottomline is I am trying to understand whether the acceptable solution sets in a and b of Figure 4 are at a comparable error level.

How was the double-difference smoothing weight selected in the MCCC technique (Eqn 2)?

Recognizing that DAS records of P-waves have relatively low SNR due to optical noise and site effect, it would still be interesting if you can return to this aspect after the results. For example, if you were to plot the raw DAS P-wave amplitudes over a 1.0 s duration window like in Figure 1f, could you see the same reversal just with very low SNR? It would be good to capture the value of information added by using this novel data processing technique, as a reference and motivation for developing future DAS data processing techniques.

Reviewer #2 (Remarks to the Author):

This manuscript principally describes an approach of using first-motion polarities measured using Distributed Acoustic Sensing (DAS) to determine earthquake focal mechanisms in areas around the Long Valley Caldera and Ridgecrest. The authors' method is intuitive, and their results are compelling. I think this manuscript would be a good fit for *Nature Communications*. I only had some trivial recommended changes, so I recommend the manuscript undergo Minor Revisions.

I think the authors did a fantastic job of describing their method and results, but I'm also familiar with the subject. It may be useful to verify that someone unfamiliar with this topic can also follow each step.

It would help to compare 1) the results produced using your method on DAS with 2) results produced by traditionally determined mechs using the existing regional network to a slightly greater extent. Were mechanisms ever compared for Ridgecrest? Simple statements like "fault plane uncertainties were reduced by $X \pm Y^\circ$..." would go a long way here.

There weren't line numbers on the manuscript that I received, so I tried copying bits of the authors' text to orient my line-by-line responses.

Abstract

"For frequent small earthquakes (magnitude < 3.5), their focal mechanisms are routinely determined using..."

I agree with what you're trying to convey, but most $M < 3.5$ earthquakes lack any determined mechanism. First motions obviously have limits too.

"which converts pre-existing telecommunication cables"

Which **can** convert. DAS doesn't have to rely on existing telecommunication infrastructure, which you give examples of later.

Introduction

"seismic monitoring of natural earthquakes"

We've used DAS to monitor induced earthquakes too. The word 'natural' could probably be dropped.

Results

"This ambiguity is corrected by determining one polarity pick"

That's relying a lot on the accuracy of a single measurement. As you discussed earlier, first-motion polarities are imperfect. It might be worth mentioning here (and/or in the Methods) that while only one polarity pick is required, but multiple picks can be used to confirm your result.

"Similarly, for the Ridgecrest DAS array"

Somewhere in this paragraph, mention that you're considering 30 earthquakes.

"we perform relative polarity measurements for 147 local earthquakes"

I thought you were using 25 earthquakes for Long Valley? Are the additional events M 0.5-1.0? It seems like you're testing your method on all 147 events, so I don't follow the earlier statement that you're testing the method on 25 events.

“Notably, for the example in Fig. 4d, a focal mechanism inversion using only DAS polarity picks can achieve the same resolution...”

I believe Fig 4d is a joint inversion. Where do you demonstrate that only DAS produces the same resolution?

Discussion

“DAS can potentially facilitate the characterization of the non-double-couple components”

This statement should be backed up with a description of how this could be done.

Figure 2

Axis labels are needed for all subfigures.

Add citation(s) for the source for the mapped faults you plotted.

Figure 3

Axis labels are needed for a-c).

Figure 4

Very cool plot!

Summary:

We thank the reviewers for their helpful comments. We have carefully considered all comments in a point-by-point manner and revised the manuscript accordingly. In the following, we mark our replies in blue color. Please note that the line numbers correspond to the document “main_diff.pdf” which tracks the changes.

Responses to reviewers:

Reviewer #1 (Remarks to the Author):

I really enjoyed reading this manuscript. It was clear and concise, and represents a step change in a fundamental earthquake parameter estimation technique – the determination of the orientation of the fault motion during an earthquake. The subject of the manuscript centers on the use of exciting new technology – the use of optical fiber networks as seismometer arrays – and develops an intricate data processing strategy to leverage this new form of data to improve the field.

There have been many journal articles about fiber-optic Distributed Acoustic Sensing (DAS) technology written in the recent years, and a few high impact articles in Nature Communications (e.g., Jousset et al., 2018; Williams et al., 2019; Munoz and Soto, 2022). The topic is high impact and publicly interesting because of how it has the potential to dramatically improve the field of earthquake science and geophysics, in general. However, many articles have only advertised “potential” impacts – this manuscript actually documents an impact. As such, I think it will be high impact among earth scientists.

Thank you for the very encouraging comments.

I have a few comments and questions for the authors, but my recommendation is to accept the manuscript with minor revisions.

Basic typos:

P3, last sentence – suggest “scatters” should be “scatterers”

Figure 2 caption – suggest “b. Earthquake location” should be “b. Earthquake locations”

Both typos have been corrected.

More significant comments:

Figure 4 is very powerful because it highlights the improvement in focal mechanism estimation enabled by DAS. Still, I was a bit confused at first because the focal plane solution in b for NC73566395 is not an acceptable solution in a.

Thank you for pointing out this. The confusion is due to a plotting issue, where only the nodal lines within 30° of the preferred solution were displayed. The preferred solution in panel b has a Kagan angle of ~39° from the preferred solution in panel a, which was outside the display range. We have now updated Figure 4 to show all accepted solutions, and the solution in b is an accepted solution in a.

As discussed in Methods the acceptable misfit ratio is 15% for inertial sensor network and 1% for DAS. For the inertial sensor network only, what is the misfit ratio a ratio of? How were the numbers 15% and 1% selected? Bottomline is I am trying to understand whether the acceptable solution sets in a and b of Figure 4 are at a comparable error level.

The misfit ratio is defined as the ratio of inconsistent polarity predictions to the total number of available polarity picks. This is calculated separately for the inertial sensor network and the DAS. We have added more descriptions in Lines 280-286.

The 15% misfit ratio threshold on the inertial sensor was selected based on the previous study (Hardebeck and Shearer 2002) and a trial-and-error approach. Specifically, for the SCSN data, a misfit ratio of 10% is allowed, which can be added by an additional 5% if the minimum misfit exceeds this value (Hardebeck and Shearer 2002). However, in the Long Valley Caldera, a 10% misfit ratio threshold frequently rejects solutions accepted by the DAS. Therefore, we increased the threshold to 15% to make the joint solutions predict consistent polarities on the inertial sensor network and DAS.

For the DAS, we chose a 1% misfit threshold as most of the joint solutions gave a misfit ratio below this value. Even if we increased the DAS misfit ratio threshold to 5%, the median misfit ratio given the joint solution is approximately 1.7%. Thus, we consider 1% as a reasonable approximation of the polarity prediction error on DAS.

In future applications of this method, these parameters can be adjusted as hyperparameters based on the quality of conventional and DAS picks, and the consistency of the joint solution between them. For example, we can perform a grid search on these hyperparameters for specific DAS cables in different regions.

How was the double-difference smoothing weight selected in the MCCC technique (Eqn 2)? The smoothing weight we used is 1. This detail has now been added to line 234. This weight was selected through trial and error. If the smoothing weight is too large, the resulting curve will be overly smooth and may not accurately represent local CC maximums. Conversely, if the smoothing weight is too small, the curve will be scattered on the global CC maximums, which could be wrong due to the cycle-skipping effect.

Recognizing that DAS records of P-waves have relatively low SNR due to optical noise and site effect, it would still be interesting if you can return to this aspect after the results. For example, if you were to plot the raw DAS P-wave amplitudes over a 1.0 s duration window like in Figure 1f, could you see the same reversal just with very low SNR? It would be good to capture the value of information added by using this novel data processing technique, as a reference and motivation for developing future DAS data processing techniques.

Thank you for this great comment. The figure below shows the inverted polarity vectors (panels a and b) and the comparison of P waveforms (panels c and d) between the two events that produced the cross-correlations shown in Figure 1f. As evident in panels a and b, the polarities between the two events are opposite on channels ~1000-2000, and the same on the remaining channels. The waveforms in panel c clearly show mirrored waveforms on channels ~1000-2000 and similar waveforms outside that range.

However, the polarity reversals of the first motions at ~1.0 seconds are not obvious. This is mainly due to the low SNR and weak sensitivity of the horizontal fiber to the vertical displacement of the P phase.

In the future, we could consider encoding these DAS polarities obtained from relative measurements into a machine learning model or use the already established DAS polarities as “templates” to infer DAS polarities of new events as discussed in Lines 191-193. We have also added more discussions in Lines 113-119.

Figure 1. Inverted polarities and waveforms of the two events that produced Fig. 1f in the main text. **a.** Inverted polarity vector of event NC73539650. **b.** Inverted polarity vector of event NC73560520. **c.** Strain rate waveforms of two events bandpass filtered from 1-10 Hz. **d.** Raw strain rate waveforms of two events.

Reviewer #2

This manuscript principally describes an approach of using first-motion polarities measured using Distributed Acoustic Sensing (DAS) to determine earthquake focal mechanisms in areas around the Long Valley Caldera and Ridgecrest. The authors' method is intuitive, and their results are compelling. I think this manuscript would be a good fit for Nature Communications. I only had some trivial recommended changes, so I recommend the manuscript undergo Minor Revisions.

I think the authors did a fantastic job of describing their method and results, but I'm also familiar with the subject. It may be useful to verify that someone unfamiliar with this topic can also follow each step.

Thank you for the very encouraging comments!

It would help to compare 1) the results produced using your method on DAS with 2) results produced by traditionally determined mechs using the existing regional network to a slightly greater extent. Were mechanisms ever compared for Ridgecrest? Simple statements like "fault plane uncertainties were reduced by $X \pm Y^\circ$..." would go a long way here.

Thanks for this great suggestion. The fault plane uncertainties in degrees can be represented by the RMS angle difference between the preferred solution and the accepted solutions. In the Long Valley test, given a misfit ratio threshold of 15% on conventional polarity picks, the RMS angle can on average be reduced by $\sim 15^\circ$ by including the DAS polarities. We have also compared the joint solution for the Ridgecrest. The RMS angle can be decreased by a similar level. We have added more discussions in Lines 147-149.

There weren't line numbers on the manuscript that I received, so I tried copying bits of the authors' text to orient my line-by-line responses.

Sorry for this inconvenience. We have added line numbers now.

Abstract

"For frequent small earthquakes (magnitude < 3.5), their focal mechanisms are routinely determined using..."

I agree with what you're trying to convey, but most $M < 3.5$ earthquakes lack any determined mechanism. First motions obviously have limits too.

We agree. We hope the DAS polarities can help to provide more constraints in the future.

"which converts pre-existing telecommunication cables"

Which *can* convert. DAS doesn't have to rely on existing telecommunication infrastructure, which you give examples of later.

That is correct, we have removed "can" in the text.

Introduction

"seismic monitoring of natural earthquakes"

We've used DAS to monitor induced earthquakes too. The word 'natural' could probably be dropped.

Definitely. We have dropped it in the text.

Results

“This ambiguity is corrected by determining one polarity pick”

That’s relying a lot on the accuracy of a single measurement. As you discussed earlier, first-motion polarities are imperfect. It might be worth mentioning here (and/or in the Methods) that while only one polarity pick is required, but multiple picks can be used to confirm your result.

We agree. Comparing multiple polarity picks from different events is indeed more robust in practice since the conventional polarities are imperfect. We have updated the text in Lines 97-99, 210-212, and 274-275 to reflect this.

“Similarly, for the Ridgecrest DAS array”

Somewhere in this paragraph, mention that you’re considering 30 earthquakes. We have now specified that we are considering 30 earthquakes in Line 110.

“we perform relative polarity measurements for 147 local earthquakes”

I thought you were using 25 earthquakes for Long Valley? Are the additional events M0.5-1.0? It seems like you’re testing your method on all 147 events, so I don’t follow the earlier statement that you’re testing the method on 25 events.

Thanks for pointing out this potential confusion. Initially, we indeed used 25 well-correlated earthquakes with catalog focal solutions in the Long Valley. This was to verify that the inverted polarities indeed correspond to the vertical displacement polarity rather than the longitudinal strain polarity.

Once we are confident in our method after the verification, we extended the analysis to more earthquakes (147 events) in that region to demonstrate the robustness of our method. The additional earthquakes cover a broader magnitude range but are not limited to M0.5-1.0. We have clarified this in Lines 123-124

“Notably, for the example in Fig. 4d, a focal mechanism inversion using only DAS polarity picks can achieve the same resolution...”

I believe Fig 4d is a joint inversion. Where do you demonstrate that only DAS produces the same resolution?

You are correct, the Fig 4d is a joint inversion. But the inversion with DAS only produces the same result. To clarify this point, we added a new panel to Fig. 4e, which demonstrates results using only DAS polarities.

Discussion

“DAS can potentially facilitate the characterization of the non-double-couple components”

This statement should be backed up with a description of how this could be done. Thanks for the suggestion. We have added more details in Lines 183-184. Specifically, we can potentially evaluate the full moment tensor solution obtained using conventional stations against the DAS polarities. For example, in waveform-based moment tensor inversion, the misfit surface of the isotropic and CLVD components can sometimes be very flat. Polarities on DAS could allow tighter constraints on searching the non-double-couple components.

Figure 2

Axis labels are needed for all subfigures.

Add citation(s) for the source for the mapped faults you plotted.

Figure 3

Axis labels are needed for a-c).

We have now updated the figures with labels added. We have also added the citation for the fault geometry and topography shown in these figures.

Figure 4

Very cool plot!

Thank you!